# Integrating tough *Antheraea pernyi* silk and strong carbon fibres for impact-critical structural composites

Kang Yang[1], Juan Guan[1,2], Keiji Numata [3], Change Wu[1], Sujun Wu[1], Zhengzhong Shao[4] & Robert O. Ritchie [5]

High stiffness and strength carbon fibres are commonly used to reinforce epoxy-resin composites. While wild *Antheraea pernyi* silk fibres exhibit high toughness originating from their α-helix/random coil conformation structures and their micro-fibre morphology, their insufficient strength and stiffness hinders them from being used in similar structural composites. In this work, we use interply hybridization of silk and carbon fibres to reinforce epoxy-matrix composites. With increased carbon fibre content, the quasi-static tensile/flexural stiffness and strength increases following the rule of mixtures while more silk fibre acts to increase ductility and impact strength. This results in a composite comprising equal volumes of carbon and silk fibres achieving an impact strength of $98\,kJ\,m^{-2}$, which is twice that of purely carbon-fibre reinforced composites ($44\,kJ\,m^{-2}$). This work shows tough natural silk fibres and strong synthetic fibres can be successfully integrated into epoxy-resin composites for tailored mechanical properties.

[1] International Research Center for Advanced Structural and Biomaterials, School of Materials Science and Engineering, Beihang University, Beijing 100191, China. [2] Beijing Advanced Innovation Center for Biomedical Engineering, Beijing 100083, China. [3] Biomacromolecules Research Team, RIKEN Center for Sustainable Resource Science, 2-1 Hirosawa, Wako-shi, Saitama 351-0198, Japan. [4] State Key Laboratory of Molecular Engineering of Polymers, Laboratory of Advanced Materials, Department of Macromolecular Science, Fudan University, Shanghai 200433, China. [5] Materials Sciences Division, Lawrence Berkeley National Laboratory and Department of Materials Science & Engineering, University of California, Berkeley, CA 94720, USA. Correspondence and requests for materials should be addressed to J.G. (email: juan.guan@buaa.edu.cn) or to R.O.R. (email: roritchie@lbl.gov)

In recent years, natural plant fibre-reinforced composites have become increasingly popular due to their green credentials and low-density characteristics[1–4]. Conversely, natural fibres from animals, such as silk (from silkworms and spiders), have been examined extensively but have found limited structural engineering applications because of insufficient strength and stiffness[5–7]. Comparing the three most popular reinforcement fibres —glass fibre, carbon fibre and flax fibre—with *Bombyx mori* (*B. mori*), *Antheraea pernyi* (*A. pernyi*) and *Nephila edulis* spider dragline silks (in Table 1), it can be concluded that silk fibres have the lowest density and the highest breaking energy among the four. In particular, the breaking energy of wild *A. pernyi* silk was found to be 150–200 MJ m$^{-3}$, which represented the highest energy absorption among all silks[8–10]. Moreover, several studies have demonstrated that *B. mori* silk fibre-reinforced plastic composites (SFRPs) could actually outperform traditionally reinforced composites as structural materials[7,11,12]. In this regard, our previous work[13] showed that a high-volume fraction of *B. mori* silk fabric, as much as 70%, could be used in epoxy-resin composites owing to its superior compressibility; furthermore, these composites displayed an enhanced impact strength that was four times higher than the unreinforced epoxy matrix. Despite their high capacity to absorb and dissipate energy, to date *A. pernyi* silk and SFRPs have not been fully exploited in polymer-matrix composites (PMCs) for structural applications. We believe a major reason for this is their insufficient strength and stiffness.

Carbon fibres, conversely, offer high modulus and tensile strength[14,15], and as such have been widely used as reinforcements for composites in aerospace and aviation fields[16–22]. However, carbon-fibre-reinforced composites (CFRP) can have low toughness and brittle mechanical properties, and this also remains a challenge for many applications[23–26]. It seems natural then to hybridize these two types of fibres. *A. pernyi* silk fibre reinforcements, with their greater ductility and superior toughness, coupled with the stronger and stiffer carbon fibres, would seem to provide a solution to the brittleness of CFRP and insufficient stiffness of SFRP. Based on this notion, our rationale for this study was to explore a hybridization of resilient-and-tough *A. pernyi* silk fibres and stiff-and-strong carbon fibres to reinforce epoxy-resin matrix composites, which we reasoned should result in an outstanding combination of mechanical properties including stiffness, strength and toughness. For example, it has been shown that the introduction of different silk fibres[27] and silks with other stiff/brittle fibres[28,29] could effectively alleviate the brittleness inherent in many epoxy-matrix PMCs. In addition, silk fibres could provide other desirable characteristics, such as lower density (the density of silk is 1300 kg m$^{-3}$, as compared with the density of 1800 kg m$^{-3}$ for carbon fibre), moderate cost and renewability.

Accordingly, in this work, we selected the tough *A. pernyi* (*Ap*) silks based on the comparative analyses of the conformation and microstructure of *Ap* silk and *B. mori* (*Bm*) silk. We then fabricated a series of hybrid-fibre-reinforced plastic composite (HFRP) laminates using *A. pernyi* silk and carbon-fibre fabrics. By varying the ratios between the stiff/strong carbon fibres and ductile/tough silk fibres, the composites were expected to display a range of mechanical properties with a variety of strength, ductility and toughness combinations. Quasi-static mechanical properties, impact properties and thermo-mechanical properties were evaluated to discern the effects of hybrid reinforcement, including composition and the stacking sequence of silk and carbon-fibre fabrics, and to understand the salient fracture mechanisms. We found that indeed the integration of tough wild *A. pernyi* silk and strong carbon fibre could complement each other to create composite materials with improved combinations of mechanical properties suitable to a wider range of structural applications.

## Results

**Characterization of *A. pernyi* and *B. mori* silk.** *A. pernyi* and *B. mori* silk from woven fabric were characterized by synchrotron wide-angle X-ray diffraction (WAXS). 1D WAXS profiles, 2D WAXS patterns and the corresponding results are shown, respectively, in Fig. 1a and Fig. 1b, c. Sharp peaks, mainly at 12.1°, 14.8°, 17.5° and 22.0°, were detected and assigned to the crystalline region of silk. The crystallinity of two silks were calculated by WAXS-peak-fitting analysis from Fig. 1a, and the fitting results were shown in Supplementary Fig. 1. The crystallinity of *A. pernyi* silk fibre was calculated as 42%, which is higher than *B. mori* silk fibre (34%). The relatively high crystallinity of *A. pernyi* silk fibre may explain its higher stiffness and strength. It is noted that the silk fibres from the fabrics contained a higher crystallinity that could originate from heat and stretching treatments during the production. The Miller indices were determined based on reported methods and shown in Fig. 1a[30]. According to the *d*-spacing values and Miller indices, the dimensions of an orthogonal unit cell were found to be *a* = 9.40 Å, *b* = 8.57 Å, *c* (fibre axis) = 6.19 Å. The crystal planes (120) and (121) in Fig. 1a indicated that β-sheet conformation is the dominant structure in both *B. mori* and *A. pernyi* fibres, but the crystal plane (200) and (202) that correlate with α-helix conformation could only be found in *A. pernyi* fibres. As shown in Fig. 1d, the helical structure can be clearly modelled from the sequence structure of *A. pernyi* fibroin. The marked difference of the conformation structure between *B. mori* and *A. pernyi* fibres may also lead to the higher toughness of *A. pernyi* fibres, as the α-helix conformation tends to offer greater elasticity and ductility than other conformations[31].

The tensile properties of *A. pernyi* and *B. mori* silk fibres were compared in an earlier study[32], and the tensile stress–strain curves of *A. pernyi* silk and carbon fibre are presented in Fig. 2e. The ultimate strengths were, respectively, 411.1 ± 80.3 and 383.5 ± 77.7 MPa for the *A. pernyi* and *B. mori* silk fibres. More importantly, the elongation and breaking energy of the *A. pernyi* fibres reached much higher values, respectively, 39.1 ± 11.3% and 97.5 ± 3.6 MJ m$^{-3}$, than the *B. mori* silk fibres. The fracture

**Table 1 Comparison of the physical, mechanical and economic properties of different fibres[8,9,51]**

| Fibre | Density (10³ kg m⁻³) | Fibre length | Tensile strength (MPa) | Tensile modulus (GPa) | Elongation at break (%) | Breaking energy (MJ m⁻³) | Biodegradable |
|---|---|---|---|---|---|---|---|
| E-glass | 2.5 | Continuous | 2000–3500 | 70–85 | 2.5–5.3 | 40–50 | No |
| Carbon | 1.8 | Continuous | 3000–4000 | 230–550 | 0.7–1.9 | 25 | No |
| Flax | 1.4 | Discrete | 800–1500 | 60–80 | 1.2–1.6 | 7–14 | Yes |
| *B. mori* silk | 1.3 | Continuous | 300–600 | 5–10 | 10–25 | 70 | Yes |
| *A. pernyi* silk | 1.3 | Continuous | 500–700 | 5–10 | 30–45 | 150 | Yes |
| Spider silk | 1.3 | Continuous | 900–1400 | 10–12 | 30–60 | 160–240 | Yes |

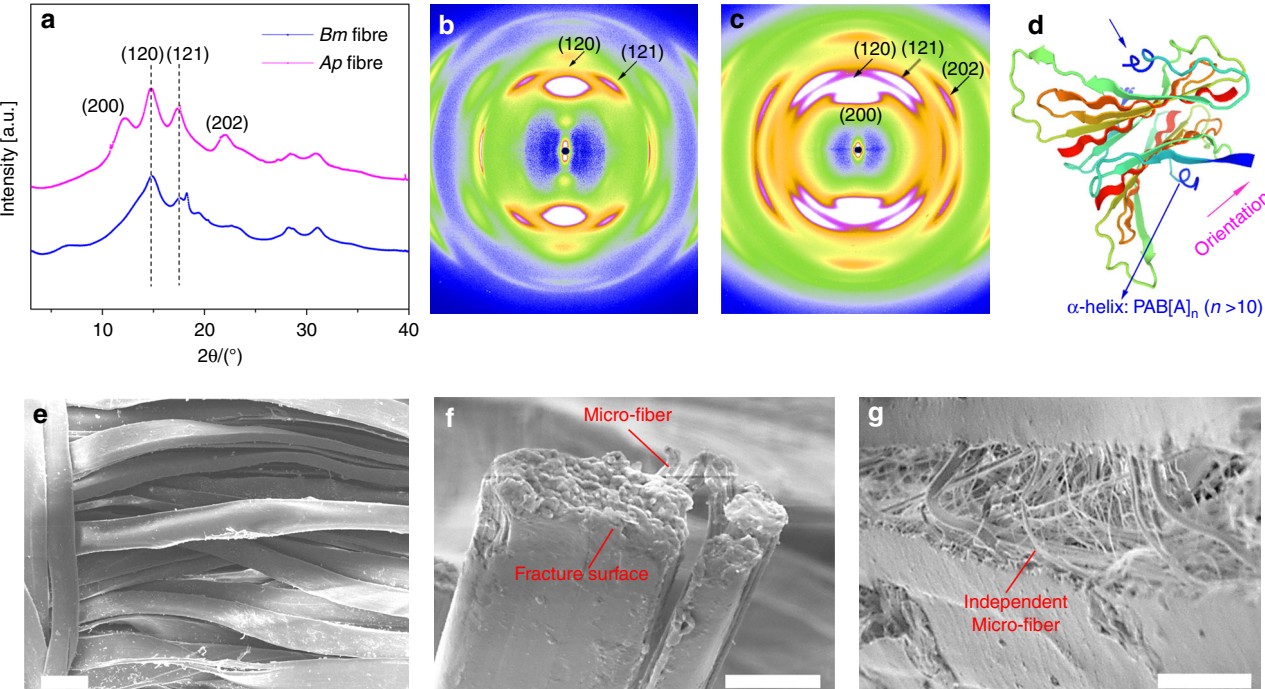

**Fig. 1** Characterization of *A. pernyi* and *B. mori* silk from woven fabric. **a** One-dimensional WAXS profile derived from the 2D WAXS patterns of silk fibres. **b**, **c** 2D WAXS patterns of a bundle of silk fibres. **b** mori silk, **c** *A. pernyi* silk. **d** Conformation structure of *A. pernyi* silk fibre. **e** Surface of *A. pernyi* silk fibre. **f** Fracture surface of *A. pernyi* silk fibre after tensile test. **g** Side of *A. pernyi* silk fibre after tensile test. Scale bars: **a** 50 μm, **b** 10 μm and **c** 20 μm

surface of a single fibre and the side fracture surface of a bunch of *A. pernyi* fibre are shown in Fig. 1f, g; the fibrillation phenomenon can be clearly observed. The balanced conformation structure and micro-fibrillar structure were critical to the unique mechanical properties especially the high ductility and toughness of the *A. pernyi* silk fibre[8]. In addition, the glass-transition temperature $T_g$ of *A. pernyi* silk was about 250 °C, i.e., much greater than the melting temperature of UHMWPE fibre as shown in Supplementary Fig. 2. Given their structure and property analyses, *A. pernyi* silk fibres, with their superior mechanical properties to the domestic *B. mori* silk, were selected to hybridize with the stiff and strong carbon fibre for our epoxy-resin composite reinforcements.

**Tensile mechanical properties of the composites**. The hybrid-fibre composites were fabricated with varying hybrid ratios of carbon fibre/silk: CFRP for 10/0; 8C2S for 8/2; 5C5S for 5/5; 2C8S for 2/8 and SFRP for 0/10, as shown in Table 2. The corresponding silk/all fibre volume ratios were, respectively, 0%, 15%, 42%, 74% and 100%. For 5C5S, three stacking sequences were prepared: 5C5S-1 with alternating carbon fibre and silk fibre layers; 5C5S-2 with silk fibre layers in the middle and 5C5S-3 with carbon-fibre layers in the middle. The cross-sectional morphology of pure SFRP can be found in Supplementary Fig. 3; it is pertinent that few voids and cracks are seen. Figure 2a–d shows the typical uniaxial tensile stress–strain curves and the derived tensile mechanical properties of the composites. With an increasing fraction of carbon fibres in HFRPs, the tensile modulus and ultimate strength increased linearly as predicted. The tensile modulus and ultimate strength of 5C5S-1 reached 39.3 GPa and 380 MPa, in comparison with 7.8 GPa and 129.3 MPa for the SFRP, thereby demonstrating the stiffening and strengthening effect of carbon-fibre reinforcements in epoxy-resin composites. On the other hand, the pure silk reinforced SFRP showed definitive yielding and far greater ductility. However, the introduction of ductile silk fibres in the HFRPs did not increase the tensile

elongation, and almost all the HFRPs still failed with tensile elongations on the order of 1%. The 5C5S hybrid composites displayed a marginal improvement in elongation from 0.91 to 1.00%, as compared with the CFRP, as shown in Supplementary Table 1. In addition, the specific tensile modulus and ultimate strength of the HFRPs could be increased to some extent considering that the density of silk (1300 kg m$^{-3}$) is lower than that of carbon fibre. For example, the specific tensile modulus for 8C2S even increased slightly compared with CFRP. For the 5C5S composites, the stacking sequence did not appear to influence the tensile properties.

To evaluate whether hybridization generates a positive or negative effect, the rule of mixtures was used[33], in the form of Eq. (1):

$$P_{\text{Hybrid}} = P_1 V_1 + P_2 V_2 \qquad (1)$$

where $P_{\text{Hybrid}}$ stands for a given property of the HFRP, $P_1$ and $P_2$ represent that property, respectively, for pure SFRP and pure CFRP, and $V_1$ and $V_2$ are the volume fractions of individual fibre reinforcements ($V_1 + V_2 = 1$). A positive hybridization effect is seen for natural composites such as bone and nacre[34,35]. Here, only the introduction of ~15 vol.% silk fibre for 8C2S resulted in a positive enhancement in the tensile strength, as shown in Fig. 2f. As suggested by Phillips[36], the hybrid effect should exist given the load-sharing assumption and experimentally observed delayed failure for hybrid FRP. We propose that the ductile and energy-absorbing *A. pernyi* silk fabric layers in 8C2S should affect the crack propagation initiated in the carbon-fibre fabrics to result in greater tensile strength.

SEM images of the tensile fracture surface of the CFRP and 5C5S-1 composites, presented in Fig. 3a, b, show that in contrast to the smooth fracture surface of the CFRP (that suggests brittle fracture is dominant), the hybrid composite 5C5S-1 exhibits more complex fracture patterns including carbon-fibre fracture, single silk fibre pull-out, silk yarn pull-out, interlaminar fracture and ductile fracture of the silk yarn. These features were also apparent for the fracture morphology of pure silk reinforced composites[13].

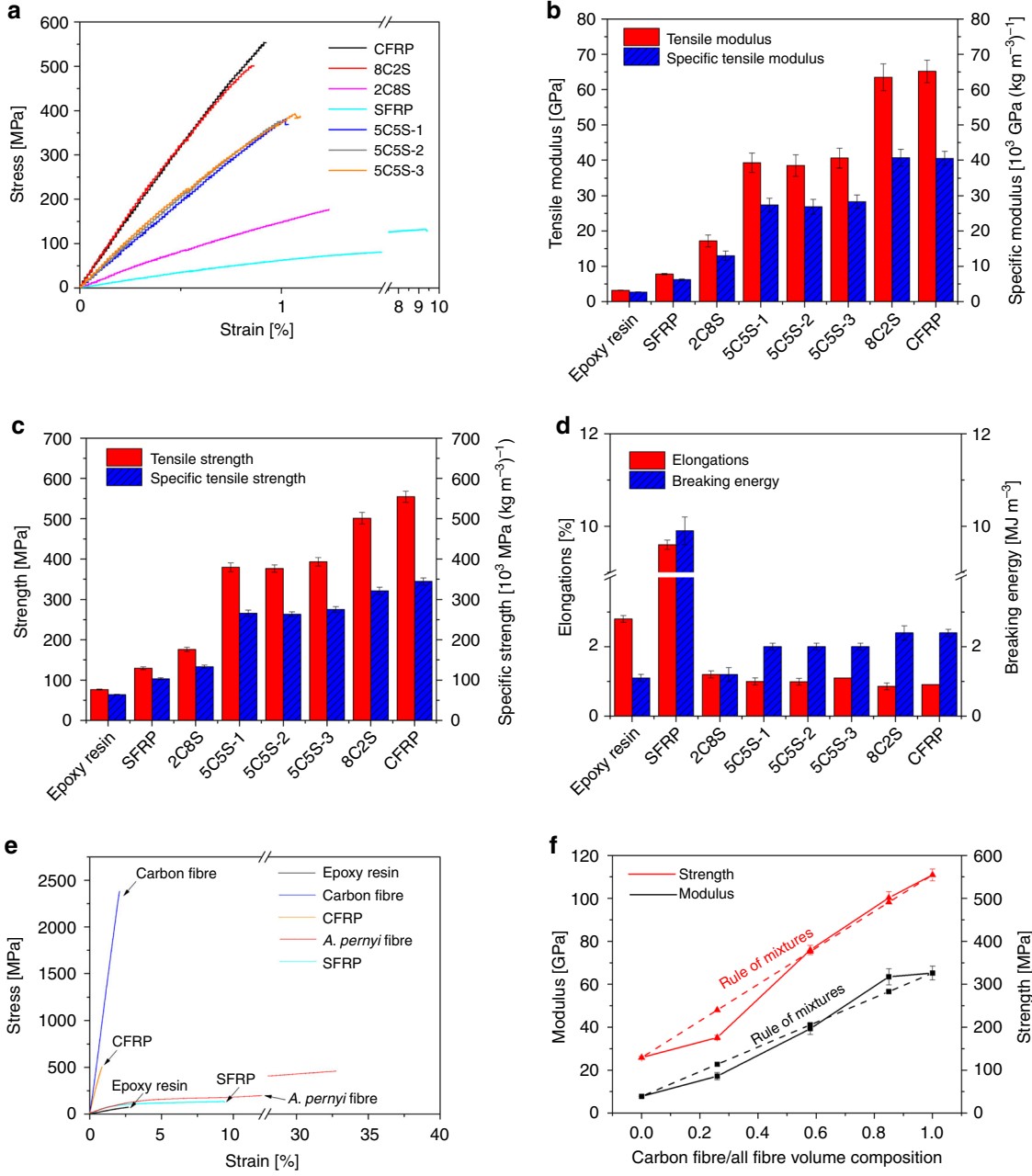

**Fig. 2** Tensile properties of composites. **a** Typical tensile engineering stress–strain curves, **b** modulus and specific tensile modulus, **c** strength and specific tensile strength, and **d** elongation and breaking energy. Note that the breaking energy is calculated as the area under the stress–strain curve and the unit is converted from MPa to MJ m$^{-3}$ for easy comparison with the literature. **e** Direct comparison of tensile stress–strain behaviour of pure epoxy resin, two reinforcement fibres, 69 vol.% CFRP and 51 vol.% SFRP. **f** Comparison of the tensile modulus and strength of the HFRP composites with increasing silk volume composition. The dashed lines indicate the expected values from the rule of mixtures (Eq. (1)). The silk volume compositions of 8C2S, 5C5S and 2C8S were, respectively, calculated as 15%, 42% and 74%. The error bars represent the standard deviation of measured means

Schematic diagrams of the distinct fracture mechanisms in CFRP and HFRP composites are also shown in Fig. 3. Overall, the hybridization of the carbon and silk fibres resulted in a predictable improvement in tensile stiffness and strength, but without any significant improvement in the tensile elongation compared with pure silk reinforcement.

**Flexural and Interlaminar shear properties of composites**. The flexural mechanical properties of the SFRP, CFRP and HFRPs are compared in Fig. 4. Different from the tensile behaviour, the flexural stress–strain curve for the HFRPs showed multiple fracture stages (Fig. 4a), whereas both the CFRP and SFRP composites showed a continuous, one-step failure. Similar to the tensile mechanical properties, a trend of increasing of flexural modulus and strength with increasing proportion of carbon fibre can be seen (Fig. 4b). The flexural modulus was increased by some 370%, respectively, from 6.8 GPa in SFRP to 31.8 GPa in 2C8S, by adding only 20% carbon-fibre reinforcement. The flexural strength of 5C5S-1 increased by ~150% to 641.3 MPa, as compared with the SFRP.

Under flexural loading, the normal and shear stress distributions in the bending mode are quite different[37], as shown in Fig. 4c. Therefore, the stacking sequence of reinforcements with varied

**Table 2 The design parameters of composite laminates with different hybrid ratios and stacking sequences**

| Laminate designation | Density ($10^3$kg m$^{-3}$) | Plies number ratio (carbon fibre/silk) | Fibre volume fraction (%) | Volume composition (carbon fibre/silk) | Stacking sequence |
|---|---|---|---|---|---|
| SFRP | 1.25 | 0/10 | 51 | 0/100 | ●●●●●●●●●● |
| 2C8S | 1.32 | 2/8 | 56 | 26/74 | ●▲●●●●●●▲● |
| 5C5S-1 | 1.43 | 5/5 | 61 | 58/42 | ▲●▲●▲●●●▲● |
| 5C5S-2 | 1.43 | 5/5 | 61 | 58/42 | ▲▲▲●●●●●▲▲ |
| 5C5S-3 | 1.43 | 5/5 | 61 | 58/42 | ●●●▲▲▲▲▲●● |
| 8C2S | 1.56 | 8/2 | 65 | 85/15 | ▲●▲▲▲▲▲▲●▲ |
| CFRP | 1.61 | 10/0 | 69 | 100/0 | ▲▲▲▲▲▲▲▲▲▲ |

Triangle and circular symbols represent carbon-fibre fabric and *A. pernyi* silk fabric, respectively

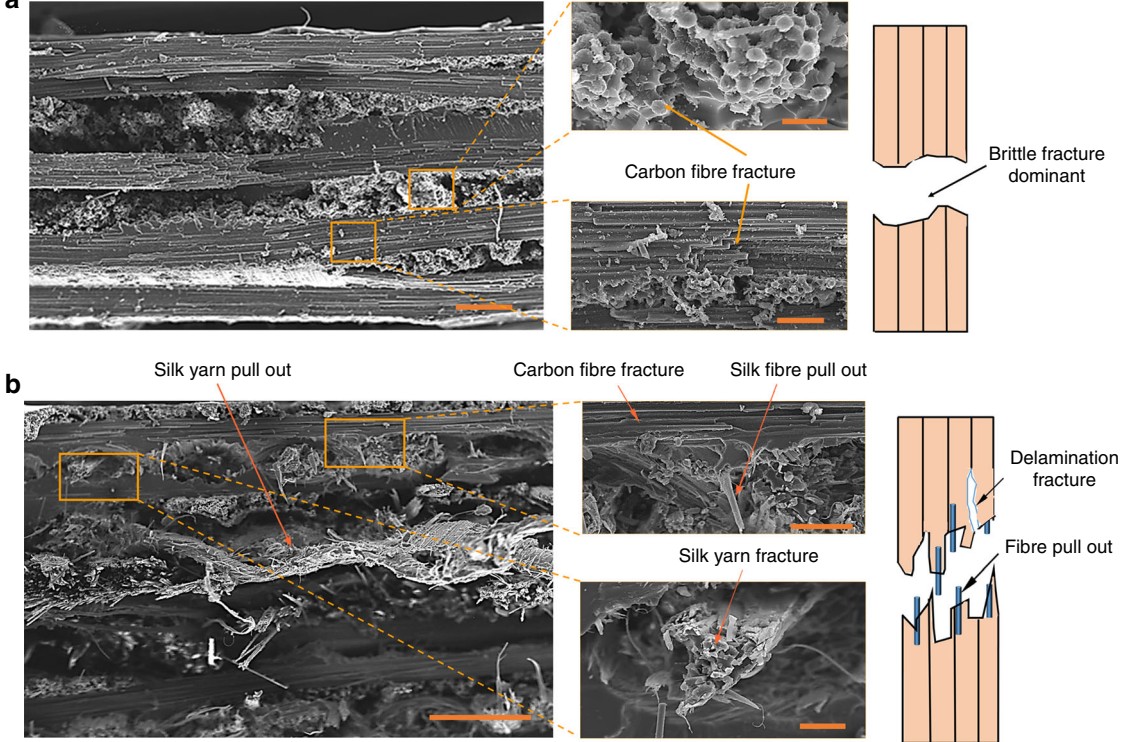

**Fig. 3** SEM image and schematic diagram of the tensile fractures. **a** CFRP, scale bars 200 μm (left panel), 20 μm (upper right panel) and 50 μm (lower right panel); **b** HFRP 5C5S-1, scale bars 500 μm (left panel), 100 μm (upper right panel) and 50 μm (lower right panel)

stiffnesses can be presumed to significantly affect the flexural properties. Comparing HFRPs 5C5S-1, 2 and 3, 5C5S-1 with alternating silk and carbon fibres showed the highest flexural modulus and strength, which suggests that stacking sequence could be tailored to provide an optimized flexural performance. With this notion, the combination of ductile silk and stiff carbon fibres could be utilized to resist both the highest normal stress on the outside and the highest shear stress on the inside, specifically under three-point bending. In addition, the flexural strength of HFRP 5C5S-1 with an alternating fabric sequence was nearly twice that of 5C5S-2 with a sandwich sequence and carbon-fibre fabric on the outer-layers, a finding that was different from earlier hybrid composite studies[38,39]. It is suggested that the higher ductility and toughness of silk fibres and the special fibre-matrix interface properties contributed to the varied flexural performance with respect to this sandwich sequence effect.

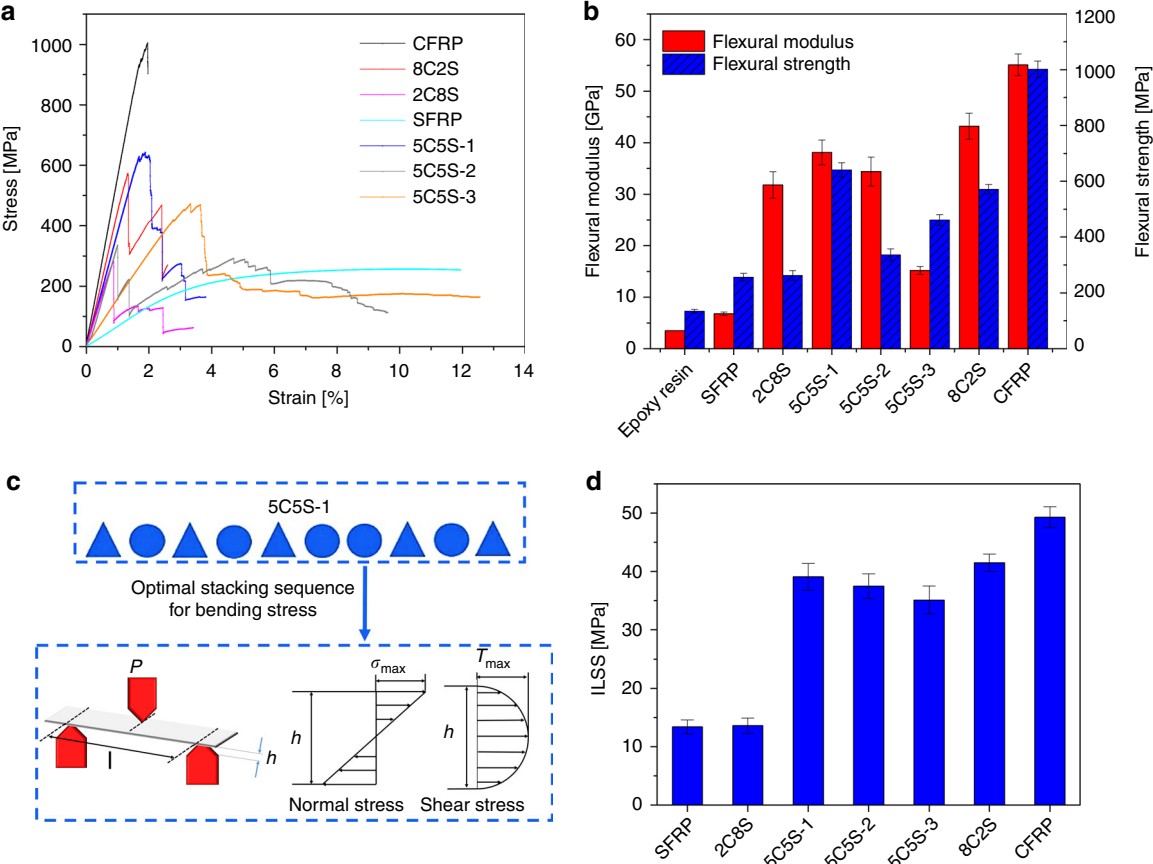

**Fig. 4** Flexural and interlaminar shear properties of composites. **a** Typical flexural stress–strain curves of CFRP, SFRP and HFRPs with various hybrid ratios and stacking sequences for the 5C5S composition. **b** Modulus and strength of CFRP, SFRP and HFRPs with various hybrid ratios and stacking sequences. **c** Stacking sequence of 5C5S-1 and distribution of normal stress and shear stress under the three-point bending mode. **d** Interlaminar shear strength (ILSS) of the CFRP, SFRP and HFRP composites with different hybrid ratios and stacking sequences. The error bars represent the standard deviation of measured means

The interlaminar shear strength (ILSS) was assessed using a three-point-bending set-up with span-to-depth ratio of 5, less than that for the above flexural test. Because of the short span, the composite would be expected to fail firstly at the interface in the middle. In other words, delamination is more likely to occur prior to other modes of failure. Figure 4d shows that the SFRP possessed the lowest ILSS; accordingly, the introduction of carbon fibres resulted in significant increase in ILSS. Specifically, 5C5S-1 and CFRP exhibited ILSS values of 39.1 MPa (290% of that for SFRP) and 49.3 MPa, respectively. Most HFRP laminates in this study were found to display more than adequate interlaminar strength for most engineering applications, as compared with the values of roughly 25 MPa that are quoted for glass-fibre and flax-fibre hybrid composites[40]. In addition, stacking sequence clearly affects the ILSS. In 5C5S-1, the greater ILSS associated with carbon fibre and the alternate sequence of carbon fibre and silk simultaneously increased the out-of-panel stiffness as well as the deformability and toughness, therefore resulted in the highest ILSS among the hybrids.

**Impact properties of composites.** Utilizing the unique mechanical properties of ductile and tough silk fibres in epoxy-resin–matrix composites has been a pivotal theme of our recent work[13,32]. In this regard, one particularly potent effect was that silk reinforcements were found to increase impact strength[13]. To examine this effect in the present hybrid-fibre composites, we utilized a modified Charpy impact experiment with unnotched test samples. The force–time curves and representative impact fracture morphologies of the various composites under study are presented in Fig. 5a, b and Supplementary Fig. 4. Compared with the CFRP, which fractured in <3 ms, hybrid-fibre composites and the SFRP displayed more plastic deformation in a prolonged fracture process of 7–10 ms. Similar to flexural deformation, the stacking sequence was also found to affect the impact performance of the 5C5S hybrid-fibre composites (Fig. 5b). In fact, the 5C5S-1 composite achieved the highest impact maximum force with more fibre fracture and pull-out, whereas the 5C5S-2 composite displayed an interesting two-peak force–time curve with more delamination between the silk layers.

The results in Fig. 5c show that the 5C5S-1 composite exhibits the highest impact strength (98 kJ m$^{-2}$), which is more than twice that of the CFRP (43 kJ m$^{-2}$). Hybridization of carbon fibres with silk fibres clearly engenders a positive effect on the impact strength of these composites. A universal force–time curve of the impact process is envisaged in Fig. 5d, which includes the initial stage of elastic resistance together with a damage propagation stage. We define here a ductility index (DI), which is the ratio of the two areas $E_p/E_i$[33,41]. $E_i$ represents the energy associated with the elastic deformation before crack initiation and $E_p$ represents plastic energy during crack/damage propagation. The silk content did not show a clear correlation with $E_i$, but stacking silk on the outside appeared to reduce $E_i$; furthermore, $E_p$ increased significantly when silk was introduced. The large

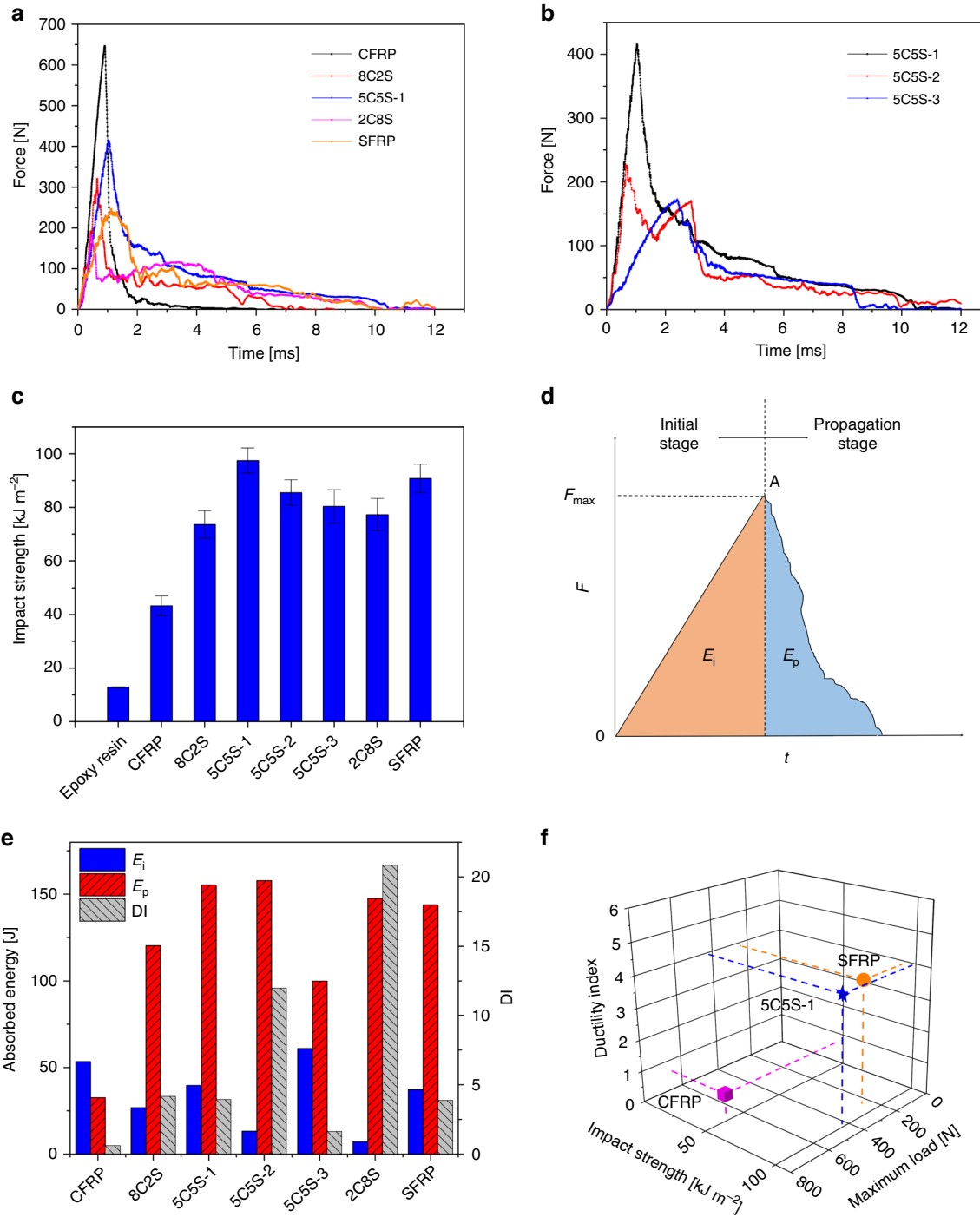

**Fig. 5** Impact properties of composites. **a** Impact force–time curves of the CFRP, SFRP and the HFRP composites with different hybrid ratios. **b** Impact force–time curves for HFRPs with different stacking sequences. **c** Impact strength of the epoxy resin as compared with that of the various CFRP, SFRP and HFRP composites with different hybrid ratios and stacking sequences. **d** Schematic force–time curve for the impact process; $E_i$ and $E_p$ represent the areas under the curve for the crack initiation and propagation stages of fracture (see text for explanation). **e** Absorbed energy $E_i$ and $E_p$, ductility index (DI) of the CFRP, SFRP and HFRPs for the impact process. **f** Comprehensive evaluation of impact properties (impact strength, maximum load and ductility index) of CFRP, Ap-SFRP and 5C5S-1. The error bars represent the standard deviation of measured means

energy-absorbing capacity of the silk fibres is associated with both elastic and plastic deformation[42,43]. In addition, further energy is absorbed from mechanisms such as fibre pull-out and interfacial debonding in SFRP[32]. In Fig. 5e, the 5C5S-1 composite can be seen to display considerable elastic and plastic energy absorption, characterized by $E_i$ and $E_p$, which resulted in the largest overall

energy absorption among all the composites in this work; this material represents an ideal balance of strength and toughness. The ductility indexes (DIs, defined as the $E_p/E_i$) of all the specimens are also shown in Fig. 5e. Interestingly, the 5C5S-2 and 2C8S composites show outstanding DIs (11.96 and 20.83), which strongly implies that stacking silk fibres on the inside with carbon

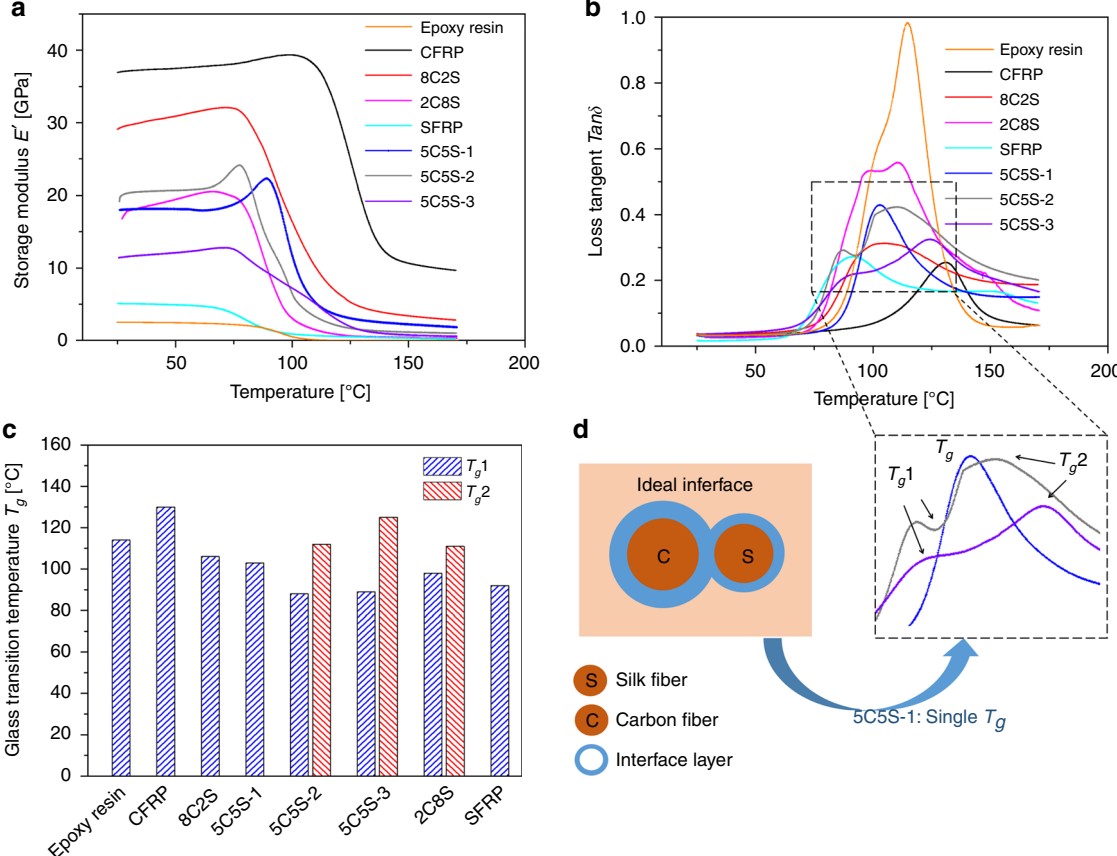

**Fig. 6** DMTA of epoxy resin and composites. **a** Dynamic storage modulus ($E'$) profiles as a function of temperature from 25 °C to 170 °C. **b** Corresponding loss tangent tan$\delta$ profiles in the same temperature range. **c** The glass-transition temperatures $T_g$ of the epoxy resin and studied composites, as defined by the peak tan$\delta$. **d** Effect of an ideal interface in 5C5S-1 in influencing the glass-transition temperatures $T_g$

fibres on the outside will effectively improve the impact ductility of these composites. To comprehensively compare the impact properties of HFRPs with CFRP and SFRP, three important evaluation indices, namely impact strength, ductility index and maximum load, are plotted on a three-axis graph in Fig. 5f. The HFRP 5C5S-1 possesses the most balanced impact performance with slightly lower maximum load but a much improved impact strength and ductility index compared with CFRP. To extend the findings, additional intra-ply hybrid fabrics were prepared; morphologies are shown in Supplementary Fig. 5. The HFRP prepared from intra-ply hybrid fabrics displayed a three-fold impact strength compared with the pure epoxy resin (see Supplementary Fig. 6). We believe that such hybrid designs could provide important guidelines for the development of superior performance in impact-critical composite materials.

**Dynamic mechanical thermal analysis of composites.** Dynamic mechanical thermal analysis (DMTA) has been widely employed for studying the molecular structures, thermo-mechanical properties and viscoelastic behaviour of polymers and polymer composites[44–47]. The dynamic storage modulus ($E'$) and the mechanical loss factor tan$\delta$ are presented in Fig. 6a, c, respectively, for the CFRP, SFRP and HFRP composites as a function of temperature from 25 to 170 °C. In this figure, the glass transition of the matrix epoxy resin can be seen to lie between 90 to 140 °C, with the value of $E'$ decreasing by two orders of magnitude simultaneously with the tan$\delta$ peak that indicated increased molecular mobility during the glass transition. Notably, the CFRP

and some HFRPs (8C2S, 5C5S-1, 5C5S-2, 5C5S-3) maintained a modulus of more than 1 GPa above the glass transition, which demonstrated some degree of a modulus enhancement effect with the introduction of carbon fibres. To demonstrate the effect of modulus enhancement through the glass transition of the matrix polymer[45], a normalized ratio or coefficient can be calculated by comparing the modulus difference of the composite and the unreinforced matrix. The effectiveness of the reinforcement can be accessed via the coefficient $C$[46], as defined in Eq. (2):

$$C = \frac{(E'_g/E'_r)_{composites}}{(E'_g/E'_r)_{resin}} \tag{2}$$

where $E'_g$ and $E'_r$ are the dynamic storage modulus, respectively, in the glassy region and rubbery region. Generally, a lower coefficient would suggest a stiffer reinforcement fibre and a stronger fibre-matrix interface. Although the high storage modulus of carbon fibres was not affected that much over the studied temperature range, the HFRPs with a higher carbon-fibre content had relatively low coefficients (i.e., 0.022 for 5C5S-1), with the CFRP, 8C2S and 5C5S-1 composites generally showing the lowest coefficients.

The glass-transition behaviour of the epoxy resin is affected by both the reinforcement hybrid ratio and the stacking sequence. In CFRP, the glass-transition temperature $T_g$ is 130 °C, whereas in SFRP the $T_g$ is the lowest at 92 °C; this is likely associated with the differing fibre-matrix interfaces for carbon fibres and silks. As described above for the interlaminar shear strength, cross-linking between the epoxy matrix and the fibres is higher for carbon

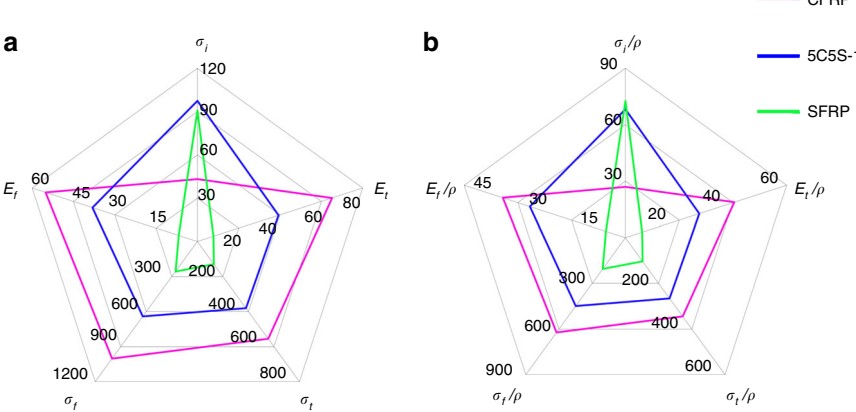

**Fig. 7** Comparative radar plots of the key mechanical properties. **a** Impact strength $\sigma_i$ (kJ m$^{-2}$), tensile modulus $E_t$ (GPa), tensile strength $\sigma_t$ (MPa), flexural modulus $E_f$ (GPa), flexural strength $\sigma_f$ (MPa). **b** Specific impact strength $10^3\sigma_i/\rho$ (kJ m$^{-2}$ (kg m$^{-3}$)$^{-1}$), specific tensile modulus $10^3E_t/\rho$ (GPa (kg m$^{-3}$)$^{-1}$), specific tensile strength $10^3\sigma_t/\rho$ (MPa (kg m$^{-3}$)$^{-1}$), specific flexural modulus $10^3E_f/\rho$ (GPa (kg m$^{-3}$)$^{-1}$), specific flexural strength $10^3\sigma_f/\rho$ (MPa (kg m$^{-3}$)$^{-1}$)

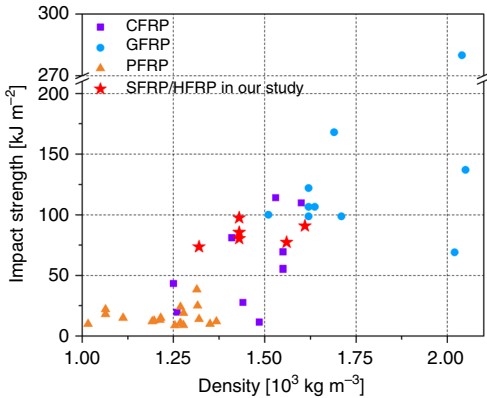

**Fig. 8** Comparison of impact strength versus density. The data are derived from the composites studied in this work and CFRP, GFRP, PFRP composites from the literature. The dataset of the properties of the composites taken from the literature, with their corresponding references, can be found in Supplementary Table 2

fibres than for silk, as an effective sizing agent on carbon fibres contains functional groups that can chemically react with epoxy groups in the matrix to result in a stronger interfacial bonding. In contrast, silk fibres were not treated chemically to improve their interfacial bonding, despite the fact that the hydroxyl and amine groups from tyrosine and lysine residues might react with epoxy groups in the matrix during processing. As a result, SFRP and HFRPs that contain dominant silk-matrix interfaces show reduced glass-transition temperatures for the epoxy resin, which suggests that the strength of the silk–epoxy interface could be further improved. On the other hand, the tan$\delta$ peaks were larger for the HFRPs than for SFRP and CFRP composites. Specifically, as the carbon-fibre content was increased, the overall tan$\delta$ peak area was reduced. It is interesting to note that the 5C5S-1 material exhibited a single tan$\delta$ peak, whereas 5C5S-2 and 5C5S-3 showed double tan$\delta$ peaks. As illustrated in Fig. 6d, different fibre/matrix interfaces could be attributed due to the various stacking sequences. 5C5S-1 contains an ideal singular interface related to the high-volume fraction and alternate carbon fibre-silk sequence. Consequently, we believe that 5C5S-1 could be a better design for a consistent glass-transition behaviour and controllable interface properties.

**Comprehensive mechanical property analysis.** For structural materials, comparative radar plots are often needed to evaluate suitable applications according to comprehensive mechanical properties. Figure 7 shows the impact strength, tensile modulus, tensile strength, flexural modulus and flexural strength and the specific (density-normalized) properties. From these plots, it is apparent that CFRP showed the highest tensile and flexural modulus and strength, whereas the composites with larger fractions of silk fibres, including SFRP, displayed twice that impact strength of the CFRP. Although not plotted in Fig. 7, the SFRP materials also possessed high tensile and flexural elongation and breaking energy due to their ability to deform plastically. Compared with SFRP, the hybrid 5C5S displayed almost the same impact strength but much higher tensile and flexural modulus and strength; compared with CFRP, the hybrid 5C5S displayed lower tensile and flexural modulus and strength but twice the impact strength. This particular set of mechanical properties of the 5C5S hybrid composite, with its silk and carbon fibres, could be ideally suited for applications where resistance to impact damage is paramount such as with wind turbine blades. Figure 8 further compares the impact strength of various composites studied in this work and from the literature. Although GFRPs show much larger impact strength, SFRPs have much lower density ($1.3 \times 10^3$ kg m$^{-3}$ for silk and $2.5$–$2.6 \times 10^3$ kg m$^{-3}$ for glass fibre). With the advantage of the low density from silk, the HFRPs in this work populate the density range of $1.3$–$1.8 \times 10^3$ kg m$^{-3}$. Given the superior impact strength of GFRPs, in the future hybrids of silk and glass fibres may be fabricated for further improved impact strength, especially with intra-ply hybrid structures.

**Other practical properties for applications.** Water absorption can be a serious issue for natural fibre-based composites. Supplementary Fig. 7 shows that the moisture pick-up for *A. pernyi* silk fibre was measured to be 7.2% after 10 days under 75% relative humidity (RH) at 25 °C, as compared with 6.1% for *B. mori* silk fibres, 12.5% for flax fibres, and 0.1% for carbon fibres. In another experiment, various composites, including pure *A. pernyi* SFRP, CFRP and HFRP with silk and carbon fibres, were immersed in water for 21 days and the mass increase was recorded (Supplementary Fig. 8). In about 5 days, the water absorption of *A. pernyi* SFRP approached a plateau of 12.5%. The HFRPs are particularly effective in reducing water absorption to merely 5%. As shown in Supplementary Fig. 9, the flexural modulus and strength of *A. pernyi* SFRP were significantly

reduced, respectively, from 5.1 GPa and 307 MPa to 1.0 GPa and 85 MPa after 21 days of water immersion. In contrast, the flexural modulus and strength of HFRP (5C5S) displayed almost an unchanged flexural modulus and strength values (5.2 GPa and 499 MPa), as compared with, respectively, values of 5.1 GPa and 503 MPa prior to water immersion. Thus, we believe that hybridization with carbon fibres could alleviate the moisture/water absorption problem for silk fibre-based composites.

Concerning the potential use of such hybrid-fibre-reinforced composites, for applications such as the blades of wind turbines, inverse creep behaviour is critical to maintain long-term functionality. Additional creep experiments using an applied tensile stress of 60 MPa for 3 days and a flexural stress of 10 MPa for 5000 min were performed. In Supplementary Fig. 10, the creep strain development of *A. pernyi* SFRP is compared with that of HFRP (5C5S). Because of the increased modulus in the HFRP, the tensile creep strain of 0.19% was smaller than that in the SFRP. Similarly, the flexural creep strain for HFRP was only 0.2% compared with 0.9% for SFRP. We conclude that the introduction of carbon fibres also can benefit the creep behaviour of silk fibre-based composites.

## Discussion

In this study, we successfully prepared hybrid-fibre fabric-reinforced epoxy-resin (HFRP) composites consisting of tough and ductile *A. pernyi* silk fibres together with strong and stiff carbon fibres. We have provided a comprehensive evaluation of the mechanical properties of such hybrid-laminate composites, including documenting the effects of hybrid ratios and stacking sequence. A higher fraction of carbon fibres clearly increases the tensile and flexural modulus and strength, whereas a higher fraction of silk fibres increases the impact strength and toughness. For carbon-fibre-reinforced composites (CFRPs), the most important improvement from silk fibre hybridization was found in the impact properties. The hybrid composite 5C5S with 42% silk fraction raised the impact strength to 98 kJ m$^{-2}$, i.e., to twice that of the CFRP. The multi-faceted fracture modes and energy absorption mechanisms for these composites are discussed, particularly the role of the plastic failure of the silk fabric. In addition, stacking alternate silk and carbon fibres in the hybrid laminates was found to result in a more balanced combination of mechanical properties and an improved silk–carbon fibre and matrix interfaces which led to an adequate interlaminar shear strength and singular glass-transition behaviour of the matrix. This work provides compelling evidence that the ductility and impact properties of CFRPs can be significantly improved for applications involving resistance to impact damage through the incorporation of tough and ductile natural silk fibres. We believe that such hybridized carbon and silk fibre-reinforced epoxy-resin materials could significantly expand the potential applications for CFRP, as well as provide a practical use of biopolymer fibre silk.

## Methods

**Materials**. Plain woven *A. pernyi* silk fabric, with an areal density of 1.35 ± 0.10 kg m$^{-2}$, was purchased from the Beijing Rui Fu Xiang Silk Store (Beijing, China); the carbon-fibre fabric, with a twill-weave structure (each yarn comprising 12k carbon fibres) with an areal density of 2.60 ± 0.10 kg m$^{-2}$, was purchased from the Weihai Development Fibre Company (Weihai, Shandong Prov., China). Plain woven silk yarns were obtained with ~110 and ~30 individual fibres, respectively, in the longitudinal and crossing directions. The average cross-sectional area of individual silk fibres was 291 μm$^2$. The density of the silk and carbon fibre was taken to be 1300 and 1800 kg m$^{-3}$, respectively[48]. The fabrics were treated at 120 °C for 24 h in an oven to remove excess moisture.

As the matrix, we chose a bisphenol epoxy resin E51, commonly used with carbon-fibre composites. A modified aromatic amine DS-300G was used as a curing agent with a mixing ratio of 100:84 (epoxy resin: curing agent) by weight. Both E51 and DS-300G were purchased from Dasen Material Science & Technology, Inc. (Tianjin, China). Additional composites described in the

Supplementary Information used a different epoxy-resin matrix from epoxy Araldite LY1564 and hardeners Aradur3486 from Huntsman Corporation (USA).

**Composite fabrication**. The silk and carbon fabrics were impregnated with epoxy resin and laid-up layer by layer prior to hot pressing. The prepreg with dimensions of 200 × 100 × 2 mm was then moulded in a hot press at a pressure of 300 kPa for 2 h at 120 °C until the curing reaction was complete. The composite laminates were fabricated with varied stacking sequence and fibre volume fraction as listed in Table 2. Each composite contained 10 layers of reinforcement fabrics. The density of all the composite laminates was measured with the Archimedes method using an electronic density balance (FA1104J); calculated density and fibre volume fractions are shown in Table 2.

**Synchrotron wide-angle X-ray scattering (WAXS) measurements**. WAXS spectra were measured at BL45XU beamline of the SPring-8 synchrotron, Harima, Japan, using X-rays with a wavelength of 0.1 nm and an energy of 12.4 keV. The sample-detector distance and exposure time were set at 187 mm and 10 s, respectively. The acquired two-dimensional (2D) scattering patterns were transferred into one-dimensional (1D) profiles via the Fit2D software[49]. The crystallinity was calculated by dividing the whole spectra area into crystalline and amorphous areas through an automatic and standard process[50]. Lorentzian functions were used for curve fitting with software Igor Pro 6.3 (WaveMetrics, Inc., Portland, OR).

**Microstructure and morphology analysis**. The microstructure and fracture morphology of the specimens before and after mechanical testing were observed in a scanning electron microscope (SEM, JEOL JSM-6010, Japan), operating at an accelerating voltage of 20 kV in the secondary electron mode. The specimens were sputter-coated with gold for 1 min using a Smart Coater (JEOL, Tokyo, Japan) prior to SEM visualization.

**Mechanical property measurements**. The tensile properties of the single *Bm* and *Ap* fibres were tested under quasi-static load control mode on a dynamic mechanical analyser (TA Instruments, Waters Ltd., DMA Q800).

Uniaxial tensile tests on dogbone-shaped specimens were conducted on an Instron 8801 screw-driven testing machine (Instron Corp., Norwood, MA, USA), at a cross-head speed of 2 mm min$^{-1}$, in accordance with Chinese Standard GB/T1040-92. Strain was measured using a strain extensometer Instron 3260 with gauge length 25 mm.

Flexural tests were performed on an Instron 5565 screw-driven testing machine, also at a cross-head speed of 2 mm min$^{-1}$, in accordance with Chinese Standard GB/T1449-2005. The specimen dimensions were 50 × 15 × 2 mm, with the span-to-depth ratio set at 12. The flexural deformation was measured as the cross-head displacement and the flexural strain was calculated using $\varepsilon_f = \frac{6Dd}{L^2}$, where $D$ represents the cross-head displacement, $d$ represents thickness of the specimen and $L$ represents the span of the specimen.

Interlaminar shear tests were conducted on the same Instron 5565 screw-driven testing machine operating at a cross-head speed of 1 mm min$^{-1}$, in accordance with the International Standard ISO 14130:1997. The specimen dimensions were 20 × 10 × 2 mm, with a span-to-depth ratio of 5.

**Impact testing**. Impact tests were carried out on an impact testing machine (model CEAST 9050, Instron Corp., Norwood, MA, USA), in accordance with the International Standard ISO179:1997 Standard. Unnotched samples were loaded flat-wise with a 2J hammer at a striking velocity of 3.8 m s$^{-1}$. The specimen dimensions were 75 × 10 × 2 mm with a span-to-depth ratio set at 20.

**Dynamic mechanical thermal analysis**. Dynamic mechanical thermal analysis (DMTA) was performed on Dynamic Mechanical Analyzer Q800 (TA Instrument, Waters Ltd.) under three-point bending mode at increasing temperature from 25 to 170 °C at a heating rate of 3 °C min$^{-1}$. The frequency and deformation strain were set at 1 Hz and 0.2%, respectively.

## Data availability

The raw data that support the findings of this study are available from the corresponding authors upon reasonable request.

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

## Acknowledgements

J.G. acknowledges Fundamental Research Funds for Central Universities. K.N. acknowledges technical support by Dr. Hiroyasu Masunaga, SPring-8, JASRI and financial support by JST ImPACT. S.J.W. and Z.Z.S. were supported by grants from the National Natural Science Foundation of China, respectively, no. 51572006, 21574024 and 21574023. R.O.R. was supported by the U.S. Department of Energy, Office of Science, Office of Basic Energy Sciences, Materials Sciences and Engineering Division under Contract No. DE-AC02-05CH11231. We thank Miss Yang Zeng, Mr. Wenhan Tian and Mr. Zihong Wu for help with fabricating additional composites and conducting additional experiments.

## Author contributions

R.O.R. and J.G. formulated the project. K.Y. carried out the experiments, analysed the data and wrote the initial paper. C.E.W. carried out some of the experiments and analysed data. K.N., S.J.W. and Z.Z.S provided experimental support and data analysis. All authors contributed to discussing the results and revising the paper.

## Additional information

**Competing interests:** The authors declare no competing interests.

