## [Transparent Peer Review File · Nature Communications]

Reviewers' comments:

Reviewer #1 (Remarks to the Author):

The manuscript entitled "Integration of tough wild *Antheraea pernyi* silk and strong carbon fibres for impact-critical structural composites" is well written and discussions and results are structured. Nonetheless, there are some major issues, some listed below, and as such I do not believe that this manuscript is suitable to be published in Nature Communication.

1- The author introduced hybridization of carbon fibre and silk fibre as the novelty of this study and proposes such hybridization for high performance composites in structural composites in aerospace and aviation (the solution for brittleness of carbon fibre). It is known that the thermal behavior of silk fibre at elevated temperature (exposure to outdoor radiation and temperature) can affect/spoil the overall performance (mechanical performance) of hybrid composites. This is why other fibres such as UHMWPE fibre is commonly used for this purpose. From performance point of view, what are the advantages of silk fibre over other types of conventional fibre such as UHMWPE fibre for hybridization with carbon fibre?

2- It is well-known that one major issue of natural fibres such as silk is moisture absorption (absorption capability of roughly between 10 to 30 weight percent). This could potentially introduce significant challenges in natural fibre reinforced epoxy composites and can be considered as a great issue regarding long term sustainability and durability. How does the author overcome this issue? How this paper claims that the toughness is merely coming from silk fibre (without considering the effect of moisture/bubble/void on toughness/ductility performance)?

3- The author used the role of mixture in this study. When it comes to equation, it is hypothesized that perfect interphase exists among constituents. However, considering silk and carbon fibre, the interfacial adhesion is a critical issue which can affect the toughness of composite through pull-out mechanism which can not only be seen for composites but also for fibres themselves. Therefore, considering the mechanical properties such as tensile stress of composites including adhesion parameter, Pukánszky's equation has been used. Can this equation or other modified equation be generalized/employed for this study?

4- The author mentioned mode I and mode III fracture. I can see mode II fracture is much more probable (in-plane shear), compared to mode III. On what basis authors ignore mode II fracture?! In reality, we can see that the tensile forces vector is not completely perpendicular to the surface plane and/or parallel to normal vector of cross section of sample. So we probably could expect in-plane shear.

5- The void coalescence can affect the ductile behavior of materials. This void coming from moisture of silk fibre and cannot be neglected. How authors distinguish the contribution of void from other toughening mechanism (such as pull-out, fibrillation) with regards to toughness?(I recommend SEM cryofracture surfaces examination (without any external load) and void content calculation can be quite good approaches to use).

6- Figure 3 (schematic), the brittle mode is considered for CFRP. It is better to say that brittle fracture is dominant. In other words, for CFRP, according to SEM images, we can see both brittle and ductile behaviors but the brittle one is much more conspicuous. For composites including fibre, the plastic deformation as well as fibre pull-out are the two most common phenomena which can lead to toughness.

7- Please provide a reference and elaborate more for this statement "Generally, a lower coefficient would suggest a stiffer reinforcement fibre and a stronger fibre-matrix interface".

8- The authors suggest the application of such material in wind turbine blades. One suggestion in this regard is to study the creep behaviour (long term behaviour) to support this. Silk fibres can demonstrate inverse creep behaviour.

9- It is strongly recommend that the authors benchmark their achievement in terms of improvement of mechanical performance with the other studies reported in literature.

Reviewer #2 (Remarks to the Author):

It is a very routine type research work. Mixed 2 different types of fibers and conducted the preliminary tests. The results are as expected according to the role of mixtures.

But the outcome numbers could be a reference values for the future researchers.

Omar Faruk

Reviewer #3 (Remarks to the Author):

The submitted manuscript demonstrates research efforts to prepare strong and tough composite laminates using both tough natural fibre (A. pernyi silk fibres) and synthetic fibres (carbon fibre). I have the following questions and suggestions:

1. The experimental design in making composite laminates with different lay-up schemes needs to be rationalised. For example, the 5C5S-2 and 5C5S-3 are not symmetrical.
2. In page 7, the authors mentioned Phillips' reference but did not elaborate the relation to this work. Please describe in detail.
3. In page 8, it is suggested to explain "type I and type III fracture modes" beforehand for a broader audience.
4. In page 9, the authors stated that "5C5S-1 with alternating silk and carbon fibres showed the highest flexural modulus and strength". This result seems to be different from literature "Hybrid composite laminates reinforced with glass/carbon woven fabrics for lightweight load bearing structures. Materials and design, 2012, vol. 36, 75-80". In that work, the conclusions are "hybrid composite laminates with 50% carbon fibre reinforcement provide the best flexural properties when the carbon layers are at the exterior, while the alternating carbon/glass lay-up provides the highest compressive strength". It used ductile glass fibre and stiff carbon fibres. Please explain.
5. Theoretical analysis seems to be necessary for validating the main experimental findings.
6. In page 14, please revise the sentence "...a sound interlaminar shear strength and singular glass transition behaviour."
7. In page 15, needs to tell the fabric structure of the carbon fibre fabric. Is it plain woven?

Reply Letter to Editor and Reviewers' Comments

We thank the editor for the precious opportunity to revise our work. Regarding the major concerns raised from the Reviewers, we have conducted additional experiments and literature survey: i) to provide the design principle for silk and carbon fibre hybrid fibre composites in this work; ii) to include a comparison of the impact strength with other fibre-reinforced composites and iii) to examine the effects of moisture and heat treatment on silk fibre-reinforced plastics and hybrid fibre-reinforced plastics. Below are the detailed point-to-point response to the Reviewers' comments, in italic text. The resulting changes made in the revised manuscript and SI are in red text.

Reviewers' comments:

Reviewer #1 (Remarks to the Author):

The manuscript entitled “Integration of tough wild *Antheraea pernyi* silk and strong carbon fibres for impact-critical structural composites” is well written and discussions and results are structured. Nonetheless, there are some major issues, some listed below, and as such I do not believe that this manuscript is suitable to be published in Nature Communication.

We thank the Reviewer for the positive comment about the structure of the manuscript. We have addressed all the major issues listed below, and we hope to convince the Reviewer that the hybrid fibre composite from silk and carbon fibre has its unique combinatorial properties and application potential.

1- The author introduced hybridization of carbon fibre and silk fibre as the novelty of this study and proposes such hybridization for high performance composites in structural composites in aerospace and aviation (the solution for brittleness of carbon fibre). It is known that the thermal behaviour of silk fibre at elevated temperature (exposure to outdoor radiation and temperature) can affect/spoil the overall performance (mechanical performance) of hybrid composites. This is why other fibres such as UHMWPE fibre is commonly used for this purpose. From performance point of view, what are the advantages of silk fibre over other types of conventional fibre such as UHMWPE fibre for hybridization with carbon fibre?

We thank the reviewer for the discussion. We had proposed that the hybridization of carbon fibre and silk fibre may be a solution for the brittleness of carbon fibres and insufficient stiffness of silk fibres for structural composite application. Accordingly, we revised the sentence on page 4 of the introduction “...provide a solution to the brittleness of CFRP and insufficient stiffness of SFRP”.

*With respect to their higher temperature characteristics, natural silks are composed of silk proteins, which have glass transition temperatures above 200°C (higher than the melting temperature 135°C for polyethylene); see “Glass transitions in native silk fibres studied by Dynamic Mechanical Thermal Analysis”, *Soft Matter* 2016, **12**, 5926, which showed *A. pernyi* silk fibre had a T_g of 250°C, much higher than that of UHMWPE. We have also conducted dynamic mechanical thermal analysis (DMTA) on both *A. pernyi* silk fibre and UHMWPE (Trevor 70 from Shandong Aidi Polymer Material Co. Ltd). The results in the Figure below showed the UHMWPE fibre could not stand temperatures above 160°C, at which temperature it rapidly softened and failed. In the revision, we have added this information in Supplementary Fig. 2.*

Silks also have a significant nitrogen and oxygen content, which could also endow them with fire retardance. The Limiting Oxygen Index (LOI) of silk is about 23.8% which is higher than the value of 17.5% for UHMWPE.

In addition to its good thermal properties, the surface chemistry or the amphiphilic nature of silk fibres means that the silk reinforcement fibre can interact with the matrix polymer to result in an enhanced fibre-matrix interface. In contrast, synthetic high-performance fibres such as Kevlar or UHMWPE are chemically inert and adhere poorly to the matrix, which often leads to inferior interfacial properties.

It is our hope these advantages of natural silks can generate new applications for polymeric composites through the hybridization of silk with carbon fibres. Nevertheless, like other polymers, silk proteins can be degraded under radiation such as UV and exposure to long-term heat or moisture. This is not an unusual problem for polymers, especially natural polymers, which would need to be managed through preventive procedures such as the introduction of protective coatings.

2- It is well-known that one major issue of natural fibres such as silk is moisture absorption (absorption capability of roughly between 10 to 30 weight percent). This could potentially introduce significant challenges in natural fibre reinforced epoxy composites and can be considered as a great issue regarding long term sustainability and durability. How does the author overcome this issue? How this paper claims that the toughness is merely coming from silk fibre (without considering the effect of moisture/bubble/void on toughness/ductility performance)?

We thank the referee for raising this issue. We agree that moisture absorption could be a pertinent issue for the application of silk-reinforced composites / SFRP. Therefore, we have conducted additional humidity conditioning experiments under relative humidities of RH ~ 75% at 25°C. The moisture absorption for the *A. pernyi* fibre reached equilibrium 7.2% after 10 days, compared to 6.1% for *B. mori* silk fibres and 12.5% for flax fibres. In another experiment, various fibre reinforced composites were immersed in water for 21 days. After 7 days, the flexural modulus and strength of *A. pernyi* SFRP was respectively reduced to 1.3 GPa and 109 MPa, and further reduced to 1.0 GPa and 85 MPa after 21 days. This deterioration in mechanical properties could not be recovered by post heat treatment.

However, by integrating the silk fibres with carbon fibres, this damaging effect could be suppressed. For example, the hybrid fibre epoxy resin composite with carbon and silk fibre reinforcements in another ongoing work showed competent flexural modulus and strength (20.2 GPa/503 MPa) after 21 days of water immersion. To overcome or mitigate the moisture/water absorption of SFRP, plying hydrophobic fabrics on the exterior or placing protective coatings may prevent water permeation; additionally, applying high pressure during processing to create dense microstructures could also reduce water permeation. We added a brief discussion in a new section 2.7 in the revised manuscript on comprehensive property evaluation of HFRPs with respect to water absorption.

“Water absorption can be a serious issue for natural fibre-based composites. Supplementary Fig. 7 shows that the moisture pick-up for *A. pernyi* silk fibre was measured to be 7.2% after 10 days under 75% relative humidity (RH) at 25°C, as compared to 6.1% for *B. mori* silk fibres, 12.5% for flax fibres, and 0.1% for carbon fibres. In another experiment, various composites including pure *A. pernyi* SFRP, CFRP and HFRP with silk and carbon fibres were immersed in water for 21 days and the mass increase was recorded (Supplementary Fig. 8). In about 5 days, the water absorption of *A. pernyi* SFRP approached a plateau of 12.5%. The

HFRPs are particularly effective in reducing water absorption to merely 5%. As shown in Supplementary Fig. 9, the flexural modulus and strength of *A. pernyi* SFRP was significantly reduced respectively from 5.1 GPa and 307 MPa to 1.0 GPa and 85 MPa after 21 days of water immersion. In contrast, the flexural modulus and strength of HFRP (5C5S) displayed almost unchanged flexural modulus and strength values (5.2 GPa and 499 MPa), as compared to respectively value of 5.1 GPa and 503 MPa prior to water immersion. Thus, it is suggested that hybridization with carbon fibres could alleviate the moisture/water absorption problem for silk fibre-based composites.”

The silk fabrics used in this work were subjected to a 120°C treatment to remove water and other volatiles. Consequently, we believe that role of water on the toughness of silk or SFRP's would be negligible. We have also tried to observe the cross-sectional morphology of the A. pernyi SFRP (Supplementary Fig. 3), and few voids/bubbles or interface defects can be seen. Although we cannot eliminate the effect, it should not be a major contribution to the overall toughness. Bearing in mind the effects on toughening from moisture and voids, the aim of this work was to demonstrate that the intrinsic toughness of silk reinforcement could lead to fracture- and impact-resistant composites.

3- The author used the role of mixture in this study. When it comes to equation, it is hypothesized that perfect interphase exists among constituents. However, considering silk and carbon fibre, the interfacial adhesion is a critical issue which can affect the toughness of composite through pull-out mechanism which can not only be seen for composites but also for fibres themselves. Therefore, considering the mechanical properties such as tensile stress of composites including adhesion parameter, Pukánszky's equation has been used. Can this equation or other modified equation be generalized/employed for this study?

We thank the reviewer for the suggestion. The rule of mixtures is a general mathematical description of a property of “mixtures” from two or more components. We resorted to this equation for its simplicity and intended to show whether a positive effect can be brought forward by hybridization. As suggested, we have also checked the literature for more theoretical models in describing the mechanical behaviour of hybrid fibre composites. However, models including Pukánszky's equation are suited only for particulate composites or composites with one continuous phase. Because predicting the stress-strain behaviour of semi-crystalline polymers such as silk remains a challenge in polymer mechanics, predicting the mechanical behaviour of silk fibre reinforced composites also remains unsolved. Actually, this is an objective of our future work. Additionally, silk and carbon fibres have distinct tensile stress-strain behaviour, and suitable models have yet to be found to describe these composites. We apologize for the lack of more generalized model for the hybrid fibre composites.

4- The author mentioned mode I and mode III fracture. I can see mode II fracture is much more probable (in-plane shear), compared to mode III. On what basis authors ignore mode II fracture?! In reality, we can see that the tensile forces vector is not completely perpendicular to the surface plane and/or parallel to normal vector of cross section of sample. So, we probably could expect in-plane shear.

We thank the reviewer for pointing this out. We apologize about the confusion here. Type I and type II and type III fractures in our original manuscript were not referring to the classical fracture mode I, II and III. In order to avoid confusion, we have omitted these statements in Figure 3 in the revision: “schematic diagrams of fracture mechanisms in CFRP and HFRP are shown on the right”.

5- The void coalescence can affect the ductile behaviour of materials. This void coming from moisture of silk fibre and cannot be neglected. How authors distinguish the contribution of void from other toughening mechanism (such as pull-out, fibrillation) with regards to toughness? (I recommend SEM cryofracture surfaces examination (without any external load) and void content calculation can be quite good approaches to use).

We thank you for the suggestion. Because A. pernyi silks and SFRP remained tough and ductile at sub-ambient temperatures (ref: Enhancing the Mechanical Toughness of Epoxy-Resin Composites Using Natural Silk Reinforcements. Sci Rep. 2017;7(1):11939.), we could not obtain a clean fracture surface to view the voids or defects. Nevertheless, we managed to view the internal morphology through grinding water-cut samples using fine sand papers. As shown in the resulting images in Supplementary Fig. 3, very few voids/bubbles were present in the composites. In addition, a vacuum treatment after lay-up of the fabrics was applied to eliminate voids and bubbles. Therefore, compared to other toughening mechanisms, we believe that contribution from voids is very limited.

6- Figure 3 (schematic), the brittle mode is considered for CFRP. It is better to say that brittle fracture is dominant. In other words, for CFRP, according to SEM images, we can see both brittle and ductile behaviors but the brittle one is much more conspicuous. For composites including fibre, the plastic deformation as well as fibre pull-out are the two most common phenomena which can lead to toughness.

Again, we thank the reviewer for the suggestion. We have corrected the wording to “that suggests brittle fracture is dominant” in CFRP. We also agree with the reviewer that plastic deformation of silk fibres and fibre pull-out are the two common mechanisms that lead to toughness. The mechanisms corresponded well to silk fibre’s ductile fracture and pull-out of silk in the schematics in Figure 3.

7- Please provide a reference and elaborate more for this statement “Generally, a lower coefficient would suggest a stiffer reinforcement fibre and a stronger fibre-matrix interface”.

We apologize for missing a reference and elaboration. In the revision, we have added the reference [46] and sentence “The effectiveness of the reinforcement can be assessed via coefficient C usually [46], shown here:

$$C = \frac{(E'_g/E'_r)_{composites}}{(E'_g/E'_r)_{resin}}$$

where E'_g and E'_r defined as dynamic storage modulus in glassy region and rubbery region.”

8- The authors suggest the application of such material in wind turbine blades. One suggestion in this regard is to study the creep behaviour (long term behaviour) to support this. Silk fibres can demonstrate inverse creep behaviour.

Actually, we have conducted creep experiments under flexural mode and tensile mode. Additional pure epoxy resin, A. pernyi SFRP and silk and carbon fibre HFRP were prepared for accelerated tensile and flexural creep tests: i) 60 MPa flexural stress at 60°C for 3 days; ii) 10 MPa tensile stress at 60°C for 5000 mins. Both the accelerated tensile and flexural creep test (Supplementary Fig. 10) showed that the addition of carbon fibre could significantly improve the creep properties. We showed that although pure silk fibres FRP experienced greater and faster creep strain development, the hybrid FRP after hybridization with carbon fibres significantly reduced the creep strain development. This hybrid FRP should be an option for wind turbine blades and further adjustments can be done through adjusting the hybrid ratio of the two fibres to achieve balanced property set for wind turbine

blades.

*We have added this discussion in a new section 2.7 “Concerning the potential use of such hybrid-fibre reinforced composites, for applications such as the blades of wind turbines, inverse creep behaviour is critical to maintain long-term functionality. Additional creep experiments using an applied tensile stress of 60 MPa for 3 days and a flexural stress of 10 MPa for 5000 mins were performed. In Supplementary Fig. 10, the creep strain development of *A. pernyi* SFRP is compared with that of HFRP (5C5S). Because of the increased modulus in the HFRP, the tensile creep strain of 0.19% was smaller than that in the SFRP. Similarly, the flexural creep strain for HFRP was only 0.2% compared to 0.9% for SFRP. We conclude that the introduction of carbon fibres also can benefit the creep behaviour of silk fibre-based composites.”*

9- It is strongly recommend that the authors benchmark their achievement in terms of improvement of mechanical performance with the other studies reported in literature.

We thank the reviewer for this excellent suggestion. Compared to GFRP with its much greater impact strength, pure SFRP has much lower density ($1.3 \times 10^3 \text{ kg.m}^{-3}$ for silk and $2.5\text{-}2.6 \times 10^3 \text{ kg.m}^{-3}$ for glass fibres). With the advantage of the low density, the hybrid FRP from silk and carbon fibre could readily populate the density range of $1.3\text{-}1.8 \times 10^3 \text{ kg.m}^{-3}$. We have added a comprehensive figure (Fig. 8) and a brief discussion in section 2.6 comparing the impact strength of various high-performance composites with regards to their densities from the literature and this work. We find that HFRPs and SFRPs displayed superior impact strength in the low-density range of $1.3\text{-}1.6 \times 10^3 \text{ kg.m}^{-3}$.

Reviewer #2 (Remarks to the Author):

It is a very routine type research work. Mixed 2 different types of fibers and conducted the preliminary tests. The results are as expected according to the role of mixtures.

But the outcome numbers could be a reference values for the future researchers.

Omar Faruk

*We thank the reviewer for the criticism. The work on hybridization of *A. pernyi* silk fibre and carbon fibre follows up well to our earlier work on silk reinforced composites (Materials & Design, Scientific Reports and Composites Part A: Applied Science and Technology). Although no one had tried to hybridize the two fibres, the more important aim was to create a novel composite with supreme impact performance, to combine the advantages including high stiffness and strength and loading taking abilities from carbon fibre and the advantages including low density and high toughness and ductility from silk fibre. We hope to convince the reviewer that the selection of fibre for hybridization in this work is no accident. Although the tensile properties of most hybrid fibre composites fit the rule of mixtures, the other focal properties especially impact properties cannot be explained by this simple rule. In addition, we have further improved the quality of this work according to the suggestions of the reviewers. Therefore, we believe the work will be of interest to the readership of this journal, and the new sets of mechanical properties from the hybrid composites will enrich the database of composite materials.*

Reviewer #3 (Remarks to the Author):

The submitted manuscript demonstrates research efforts to prepare strong and tough composite laminates using both tough natural fibre (A. pernyi silk fibres) and synthetic fibres (carbon fibre). I have the following questions and suggestions:

1. The experimental design in making composite laminates with different lay-up schemes needs to be rationalised. For example, the 5C5S-2 and 5C5S-3 are not symmetrical.

We thank the reviewer for this question. In this work, we applied 10 layers of fabrics for composite fabrication. For the 2C8S and 8C2S compositions, the minor fibre layers were placed next to out-layers and symmetrically on both sides. For the 5C5S composition, 5 layers of carbon fibre fabric and 5 layers of silk fabric were placed in three configurations or lay-ups: alternating silk and carbon fibre layers and placing carbon fibre layers as the out-layers for 5C5S-1; sandwich structures with carbon fibre layers as out-layers and silk layers in the middle for 5C5S-2; sandwich structure with silk fibre layers as out-layers and carbon fibre layers in the middle for 5C5S-3. As also suggested by Reviewer 1, we have added the explanation for the rationalization of the laminate lay-up design in the revised text.

2. In page 7, the authors mentioned Phillips' reference but did not elaborate the relation to this work. Please describe in detail.

Thanks for the suggestion. In ref. 36 authored by Phillips, hybrid effects using an example of hybrid FRP from carbon fibre and glass fibre in the same vinyl ester resin matrix were demonstrated. We have revised this sentence in the revision: "As suggested by Phillips [36], the hybrid effect should exist given the load-sharing assumption and experimentally observed delayed failure for hybrid FRP. We propose that the ductile and "energy absorbing" A. pernyi silk fabric layers in 8C2S should affect the crack propagation initiated in the carbon fibre fabrics to result in greater tensile strength."

3. In page 8, it is suggested to explain "type I and type III fracture modes" beforehand for a broader audience.

We thank you for the suggestion. We are sorry for missing this information. Type I, type II and type III fractures in our initial manuscript were referring to brittle fracture, mixed mode fracture and ductile fracture. They were different from the fracture Mode I, II and III corresponding to crack opening fracture, out-of-plane shear fracture and in-plane shear fracture. As noted above, to avoid confusion, we have changed this in the figure and text to "schematics of fracture and toughening mechanisms".

4. In page 9, the authors stated that "5C5S-1 with alternating silk and carbon fibres showed the highest flexural modulus and strength". This result seems to be different from literature "Hybrid composite laminates reinforced with glass/carbon woven fabrics for lightweight load bearing structures. Materials and design, 2012, vol. 36, 75-80". In that work, the conclusions are "hybrid composite laminates with 50% carbon fibre reinforcement provide the best flexural properties when the carbon layers are at the exterior, while the alternating carbon/glass lay-up provides the highest compressive strength". It used ductile glass fibre and stiff carbon fibres. Please explain.

Thanks for the discussion. The work on hybrid composite from ductile glass fibre and stiff carbon fibre did show that in the 50% carbon fibre composition when the carbon fibre layers are placed at the exterior, the flexural properties are the best. In our work, when the carbon fibre layers and the silk fibre layers are alternating, the flexural properties are the best. Moreover, the impact strength of this type of lay-up is the highest. There may be two reasons for these observations. Firstly, the A. pernyi silk fibre is a more ductile fibre than the glass fibre. The tensile elongation of A. pernyi silk fibre is ~40%, much greater than that of glass

fibre ~6%. The larger plastic deformation of *A. pernyi* silk could promote different effects on the flexural mechanical properties. Secondly, as we discussed in the dynamic mechanical thermal analysis section, the interface property of the 5C5S-1 composite with alternating silk and carbon fibres was better than the other sandwich structured composites 5C5S-2 and 5C5S-3. The closely packed carbon fibre layer and silk fibre layer likely form an ideal interface matrix phase. We believe that this is the main reason for the superior flexural and impact properties of 5C5S-1. In the revision, we have added this explanation in section 2.3. “In addition, the flexural strength of HFRP 5C5S-1 with an alternating fabric sequence was nearly twice that of 5C5S-2 with a sandwich sequence and carbon fibre fabric on the outer-layers, a finding that was different from earlier hybrid composite studies [38,39]. It is suggested that the higher ductility and toughness of silk fibres and the special fibre-matrix interface properties contributed to the varied flexural performance with respect to this sandwich sequence effect.”

5. Theoretical analysis seems to be necessary for validating the main experimental findings.

In response to this suggestion, we have added more discussion on the overall properties of silk fibre and hybrid fibre-based composites to include the moisture effect and creep behaviours in a new section 2.7, and a comparison of the impact properties of our composites to various composites in section 2.6. As this work was an experimental study, theoretical analysis or modelling was outside the scope of the program. As we explained to Reviewer 1, predicting the stress-strain behaviour of ductile silks and silk-based composites remains a problem. However, in our future studies, we intend to exploit finite element analysis and modelling to further validate our experimental findings on these materials.

6. In page 14, please revise the sentence “...a sound interlaminar shear strength and singular glass transition behaviour.”

Thanks for the suggestion. We have replaced the word “sound” to “adequate” (39 MPa ILSS) and revised this sentence to “...properties and an improved silk-carbon fibre and matrix interfaces which led to an adequate interlaminar shear strength and singular glass transition behaviour of the matrix.”

7. In page 15, needs to tell the fabric structure of the carbon fibre fabric. Is it plain woven?

*Thanks for the suggestion. The *A. pernyi* silk fabric was plain woven whereas the carbon fibre fabric was twill-weaved. We have added this information in the materials section 4.1 with the phrase “with a twill-weave structure...”*

REVIEWERS' COMMENTS:

Reviewer #1 (Remarks to the Author):

I carefully reviewed authors response to reviewers comments and the implemented changes to the manuscript to address those comments. I am in particular impressed by how well authors responded to my comments (reviewer 1 in the first review) and as such I recommend publishing this manuscript in its current format.

Associate Professor Minoos Naebe

Reviewer #3 (Remarks to the Author):

Thank you for the revisions made according to the comments. The manuscript is satisfactory for publication.

Jin Zhang

REVIEWERS' COMMENTS:

Reviewer #1

(Remarks to the Author): I carefully reviewed authors response to reviewers comments and the implemented changes to the manuscript to address those comments. I am in particular impressed by how well authors responded to my comments (reviewer 1 in the first review) and as such I recommend publishing this manuscript in its current format.

Associate Professor Mino Naebe

Response: We appreciate the excellent comments and suggestions from Prof. Mino Naebe in the first round and the careful check of our revisions.

Reviewer #3

(Remarks to the Author): Thank you for the revisions made according to the comments. The manuscript is satisfactory for publication.

Jin Zhang

Response: Thanks to Dr Jin Zhang for the recommendation.